# OpenReview forum: "ShiQ: Bringing back Bellman to LLMs"
_NeurIPS.cc/2025/Conference — NeurIPS 2025 poster_

### Official Review · Reviewer_WKHJ · 2025-06-30

**Clarity:** 3
**Significance:** 3
**Originality:** 2
**Rating:** 5
**Confidence:** 4

**Summary:**

The authors consider the backward recursion for computation of KL-regularized state-action function (Theorem 1). They then consider the minimization of KL-regularized Bellman error and provide a more tractable shifted-Q formulation of Bellman error (Theorem 2).  Finally, they propose an algorithm that supports off-policy, token-based KL regularized Q learning for LLM training and evaluate performance in various benchmarks.

**Questions:**

1.  I would appreciate the  authors comment on the merits of the proposed approach given the lack of token-based rewards. Perhaps the proposed approach provides a principled approach for credit assignment. If that is the case the right benchmarking comparison should be other approaches credit assignment rather than sequence-based LLM training methods.

2. Can the authors elaborate on the computational burden imposed by the length limit T_max ? Could there be ways to mitigate this ?

**Ethical Concerns:**

["NO or VERY MINOR ethics concerns only"]

**Final Justification:**

I have read the author rebuttal and considered all raised points., I have engaged in discussions and responded to authors., I have filled in the "Final Justification" text box and updated "Rating" accordingly (before Aug 13) that will become visible to authors once decisions are released., I understand that Area Chairs will be able to flag up Insufficient Reviews during the Reviewer-AC Discussions and shortly after to catch any irresponsible, insufficient or problematic behavior. Area Chairs will be also able to flag up during Metareview grossly irresponsible reviewers (including but not limited to possibly LLM-generated reviews)., I understand my Review and my conduct are subject to Responsible Reviewing initiative, including the desk rejection of my co-authored papers for grossly irresponsible behaviors. https://blog.neurips.cc/2025/05/02/responsible-reviewing-initiative-for-neurips-2025/

**Quality:**

3

**Strengths And Weaknesses:**

Strengths

The paper is well-written, and the results seem correct. Developing a backward recursion scheme for KL-regularized Q learning is not necessarily a significant contribution. Instead, I see the main contribution of the paper in the adaptation of such framework to LLM training.

Weaknesses

- The  authors do not expand upon how a reliable token-based reward model can be assumed when in practice this token-based feedback is generally not available.
- It appears the computational effort of the shifted- Q algorithm grows exponentially in T_max. This may restrict its practical application to short sequences.

---

> ### Author Rebuttal · Authors · 2025-07-28
>
> We extend our gratitude to the reviewer for their valuable time and insightful comments and questions.
>
>
> Weaknesses
>
>
>     Developing a backward recursion scheme for KL regularised Q learning is not necessary a significant contribution.
>
> The proof to our knowledge has not been done with $T_{max}$ as a random variable, although the proofs are similar with the fixed horizon (or max sequence lenght).
>
>     The authors do not expand upon how a reliable token-based reward model can be assumed when in practice this token-based feedback is generally not available.
>
>     and
>
>      q1) I would appreciate the authors comment on the merits of the proposed approach given the lack of token-based rewards. Perhaps the proposed approach provides a principled approach for credit assignment. If that is the case the right benchmarking comparison should be other approaches credit assignment rather than sequence-based LLM training methods.
>
>
> Even though we do not have access to token-level rewards, thanks to the multi-turn aspect of our algorithm, we manage to propagate information and obtain some signal. For example, in HH, the reward model only provides a reward for the entire trajectory, and it works well. Finally, even if there are no datasets with rewards per token, we have multi-step rewards in BFCL, which allows for granularity in ShiQ and enables us to benefit from it.
>
>
>     It appears the computational effort of the shifted- Q algorithm grows exponentially in T_max. This may restrict its practical application to short sequences.
>
>     and
>
>       q2) Can the authors elaborate on the computational burden imposed by the length limit T_max ? Could there be ways to mitigate this ?
>
>
>
> We disagree that ShiQ grows exponentially with $T_{max}$. If you are referring to Equation 10 with the two sums, it is possible to avoid an explosion in the implementation.
>
>
> Our loss is an external sum of cumulative sums. In the first pass, we calculate the internal cumulative sum starting from the index at $T_{max}$. Each term, when traversing the index in reverse, is simply the previous term plus one term. At the end, we sum all the terms. Therefore, we do not have an explosion of loss terms, and the complexity is linear in $T_{max}$, not exponential (for calculating the loss terms, assuming they are already computed). However, we do pay in memory because we need to store the terms, but again, there is no exponential complexity!

---

### Official Review · Reviewer_VzGK · 2025-07-02

**Clarity:** 2
**Significance:** 2
**Originality:** 3
**Rating:** 4
**Confidence:** 4

**Summary:**

This paper proposes an offline reinforcement learning algorithm called ShiQ (Shifted-Q) for fine-tuning LLMs. The core contribution is deriving theoretically grounded loss functions from Bellman equations to adapt Q-learning methods to LLMs, addressing challenges like sample inefficiency and the high computational cost of sampling with LLMs. The authors claim that ShiQ supports off-policy, token-wise learning and is simple to implement. They evaluate ShiQ on synthetic data and real-world benchmarks, including UltraFeedback and BFCL-V3, in both single-turn and multi-turn LLM settings.

**Questions:**

1. From the three sub-methods proposed (Easing sampling, Improved initialization, Multi-step extension), which one do you believe contributes most significantly to ShiQ's performance, and why?
2. Beyond empirical performance, what are the primary theoretical advantages of your Q-learning-based approach compared to traditional policy-gradient methods in the context of LLM fine-tuning?
3. How would you address the scaling challenges if ShiQ were to be applied to significantly larger LLMs and datasets? What potential issues might arise in a large-scale deployment, and how would you mitigate them?
4. Is ShiQ more suitable for preference learning tasks (like aligning with human preferences) or for fine-tuning LLMs on structured reasoning tasks, similar to chain-of-thought (CoT) training in DeepSeek-R1?

**Ethical Concerns:**

["NO or VERY MINOR ethics concerns only"]

**Final Justification:**

Thank the authors for their detailed explanation. Most of my concerns have been addressed. However, I still doubt the significance of the proposed method. Nevertheless, considering the contribution of the proposed method, I would like to increase my score.

**Limitations:**

Yes, the authors have adequately addressed the limitations.

**Paper Formatting Concerns:**

No formatting concerns

**Quality:**

2

**Strengths And Weaknesses:**

**Strengths:**

1. The paper tackles a crucial problem in LLM fine-tuning, specifically the sample complexity associated with reinforcement learning. Efficiently fine-tuning LLMs with RL is a significant challenge.
2. The paper provides a comprehensive review of existing literature, covering various RL fine-tuning approaches for LLMs.
3. The motivation behind adapting Q-learning to LLMs, highlighting its potential for sample efficiency and offline learning, is well-articulated and reasonable.
4. The approach of deriving theoretically grounded loss functions from Bellman equations to adapt Q-learning to LLMs, especially with the proposed transformations (easing sampling, improved initialization, multi-step extension), shows a degree of novelty.

**Weaknesses:**

1. While the method appears theoretically sound, the derivation involves multiple transformations and notations that make it overly complex. It's challenging to grasp the full intuition behind each step.
2. The empirical evaluations are a major weakness.
    * Most experiments are on toy datasets, which doesn't fully demonstrate the method's effectiveness in real-world, large-scale LLM scenarios.
    * Crucially, the experimental figures lack error bars, making it difficult to assess the statistical significance and robustness of the results.
3. The significance of the results is questionable. In Figures 3 and 4, ShiQ's performance is comparable to, or sometimes even slightly worse than, the baseline CoPG. The benefits claimed in the introduction (e.g., sample efficiency) are not clearly demonstrated by the empirical results.
4. The paper's clarity suffers from inconsistent notation. It mixes RL and LLM notations throughout, which is confusing. A unified and simplified notation would greatly improve readability.

---

> ### Author Rebuttal · Authors · 2025-07-28
>
> We would like to express our sincere gratitude to the reviewer for their valuable time, insightful comments, and thoughtful questions.
> Weakness: (4 points)
>
>     w1) While the method appears theoretically sound, the derivation involves multiple transformations and notations that make it overly complex. It's challenging to grasp the full intuition behind each step.
>
>
> Thanks for your remark. We tried to keep the notations as simple as possible for the RL side. We also included graphs to explain our approach just before Section 2.1, in the main text, and all proofs are self-contained. Each paragraph first provides the intuition behind our approach and then attempts to be more precise about exactly what we are doing. We would appreciate advice on any sections that may be complicated for the reader, to improve the readability of our paper
>
>
>
>     w2)The empirical evaluations are a major weakness.
>     Most experiments are on toy datasets, which doesn't fully demonstrate the method's effectiveness in real-world, large-scale LLM scenarios.
>
> We kindly disagree on this point. There is extensive evaluations with lots of ablations for our method.
> The toy experiments are for giving intuitions. ( ie bandit and MPD setting) Then, there is experiments on 3 differents environments with HH, UltraFeedback and BFCL-v3 which is multi-turn setting so we do not agree that is not large scale LLM scenario, especially for BFCL-V3.
>
>
>     Crucially, the experimental figures lack error bars, making it difficult to assess the statistical significance and robustness of the results.
>
>
> We will definitely add it for bandits and MDPs. For LLMs, a seed is very costly, and there are only a few papers that can afford to include error bars with agentic benchmarks.
>
>     W3)The significance of the results is questionable. In Figures 3 and 4, ShiQ's performance is comparable to, or sometimes even slightly worse than, the baseline CoPG. The benefits claimed in the introduction (e.g., sample efficiency) are not clearly demonstrated by the empirical results.
>
>
> CoPG is generally a strong baseline. Then, in Figure 4, under the multi-turn setting, our method, ShiQ, clearly outperforms. Moreover, in terms of the Pareto front, our method is superior in both Figure 3 (single-turn setting) and Figure 4 (multi-turn setting).
>
>
>     W3)The paper's clarity suffers from inconsistent notation. It mixes RL and LLM notations throughout, which is confusing. A unified and simplified notation would greatly improve readability.
>
>
> First, we introduce a rigorous notation that accounts for dense rewards at each step, similar to reinforcement learning. These notations are necessary to derive all the proofs. Then, in a second step, we simplify the notations for practitioners so they can easily implement the loss. The notations are simplified insofar as the reward is sparse and only occurs at the end of the sequence. At no point notations are inconsistent. Moreover, we always clarify the connection between the two notations.
>
>
>     From the three sub-methods proposed (Easing sampling, Improved initialization, Multi-step extension), which one do you believe contributes most significantly to ShiQ's performance, and why?
>
>
> All points in our method are important, as emphasized in the appendix with the ablation study on the initialization shift and the multi-step aspect when the reward is sparse (see Figures 5 and 6 in the appendix).
>
>
> The easy sampling aspect is very important because otherwise,  we would need to load two models during sampling (and having same temperature), which is extremely costly when scaling up, and this is not acceptable. The idea of the paper is to have the LLM policy as the softmax of the logits, so a simple formulation at the infrastructure level. As highlighted in Figure 5 of the appendix, Shiq no ms (for no multistep) has no signal and does not increase the reward, so it is very important in an environment with sparse rewards. Perhaps the least important is the shift, as we can still achieve performance, but we always have this initialization problem for learning.
>
>
>     Beyond empirical performance, what are the primary theoretical advantages of your Q-learning-based approach compared to traditional policy-gradient methods in the context of LLM fine-tuning?
>
>
> The advantages are twofold. First, compared to classical policy gradients, our method has theoretical convergence guarantees to the best policy in the offline setting. Second, it allows for fine-grained allocation, which is not the case with classical policy gradients
>
>
>     How would you address the scaling challenges if ShiQ were to be applied to significantly larger LLMs and datasets? What potential issues might arise in a large-scale deployment, and how would you mitigate them?
>
>
>
> The whole idea of the paper is to avoid excessive computational burden, such as a value network, etc. Compared to a policy gradient, it typically fits on the same GPU partition because we only load one LLM without a target network. Therefore, our method scales well with the size of datasets, at least as well as methods like GRPO, CoPG, etc.
>
> From a computational perspective, the loss is similar to cross-entropy when you think about it. Instead of a single term over the trajectory, we have a sum of cumulative inverse sums, which is fundamentally like cross-entropy. If implemented correctly, we calculate the cumulative sum first, starting from the term with the least value, i.e., at index T_max, and sum at the end. This does not cause the loss calculation to explode exponentially with $T_{max}$ but is $O(T_{max})$ for computing the loss terms, assuming the terms are already calculated.
>
>
> Furthermore, when comparing to policy gradients, we sample multiple times to obtain a baseline in policy gradients, which can be very costly. This is not the case in our method.
>
>     Is ShiQ more suitable for preference learning tasks (like aligning with human preferences) or for fine-tuning LLMs on structured reasoning tasks, similar to chain-of-thought (CoT) training in DeepSeek-R1?
>
> The question for us is not about the number of turns but rather depends on the granularity of the signal. If, in the CoT, we have information for each segment with a reward, then ShiQ makes complete sense. If we are simply doing reference learning in a single turn, then a bandit algorithm like DPO is better suited for simple LLMs, as there is no reason for it not to work well. If we have intermediate reward information, such as in code with passing tests, then ShiQ has a better chance of functioning effectively.

---

### Official Review · Reviewer_CDVa · 2025-07-02

**Clarity:** 3
**Significance:** 2
**Originality:** 3
**Rating:** 4
**Confidence:** 3

**Summary:**

The author starts from the Bellman equation of soft Q-learning and introduces three improvements: reducing the size of the final policy, ensuring zero loss at initialization when the target is a reference policy, and extending the optimization objective to multi-turn scenarios, ultimately deriving the ShiQ optimization formulation. Experiments were conducted on toy examples, single-turn tasks (HH and UltraFeedback), and multi-turn tasks (BFCL-V3). The results show that ShiQ performs comparably to alignment methods like CoPG in single-turn tasks and outperforms baseline methods in BFCL.

**Questions:**

1. For the implementation details, during optimization, are $v_{\ell}$ and $\pi_{\ell}$ in Eq.10 updated simultaneously? Is stop-gradient applied? If the gradient of ShiQ with respect to $\pi$ can be provided in the paper, it may be more convenient to understand the optimization objective.

2. In the multi-turn experiments, does DRO use single-step rewards or multi-step returns? If DRO is modified to use multi-step returns, would it be equivalent to ShiQ (ignoring value modeling differences)? What would the empirical results show?

3. For smaller models (e.g., 7B/4B), where memory constraints are less severe, would using an additional value network, a tiny shared-parameter network, or the proposed value difference yield different results? Is an extra value network still necessary for optimal performance?

Minor Issue:
The subscripts in Equation 9 are incorrect？

**Ethical Concerns:**

["NO or VERY MINOR ethics concerns only"]

**Final Justification:**

My assessment of the strengths and weaknesses remains largely consistent with my initial evaluation before the rebuttal. During the rebuttal, the author clarified some implementation details of ShiQ and DRO, but my key concerns persist:

While ShiQ demonstrates promising results, it would be valuable to investigate whether simpler heuristics could achieve comparable performance.

Regarding the value network ablation study, even a small-scale comparison (feasible using existing frameworks for 7B-scale models) would provide stronger validation.

Since the author did not present experimental evidence addressing these points, I am maintaining my original rating.

**Limitations:**

Yes

**Quality:**

3

**Strengths And Weaknesses:**

Strengths:
1. The formulation of ShiQ is theoretically grounded, starting from soft Q-learning.
2. Ablation studies are provided for the initialization trick and multi-turn extension, demonstrating their effectiveness.

Weaknesses:

1. ShiQ's final formulation closely resembles DRO [1], without significant improvements.

The author claims two main improvements over DRO:

(1) Token-level and multi-step rewards: ShiQ handles these, but DRO could also be trivially extended by replacing single-step rewards with multi-step returns.

(2) Memory-friendly value modeling: ShiQ replaces DRO's separate value network with the proposed value difference method, but no experiments compare the performance impact of value difference versus an additional value network.

[1] Richemond, Pierre Harvey, et al. "Offline regularised reinforcement learning for large language models alignment." arXiv preprint arXiv:2405.19107 (2024).

---

> ### Author Rebuttal · Authors · 2025-07-28
>
> We thank the reviewer for their time and for their pertinent remarks and questions.
>
>
>     ShiQ's final formulation closely resembles DRO [1], without significant improvements.
>     The author claims two main improvements over DRO:
>
>     (1) Token-level and multi-step rewards: ShiQ handles these, but DRO could also be trivially extended by replacing single-step rewards with multi-step returns.
>
>
> We disagree that ShiQ is similar to DRO for several reasons. DRO operates in a bandit setting and is single-step. In our paper, we use MDPs, and adapting from a bandit to an MDP DRO algorithm is not straightforward. Furthermore, we do not agree that there is no significant improvement, especially in the multi-turn case (see Figures 3 and 4).
>
>
>     (2) Memory-friendly value modeling: ShiQ replaces DRO's separate value network with the proposed value difference method, but no experiments compare the performance impact of value difference versus an additional value network.
>
>     q3) For smaller models (e.g., 7B/4B), where memory constraints are less severe, would using an additional value network, a tiny shared-parameter network, or the proposed value difference yield different results? Is an extra value network still necessary for optimal performance?
>
> Multiplying large networks like LLMs is undesirable, as it strains hardware resources and leads to wasteful memory consumption. Additionally, this approach is challenging to implement, as noted in DRO, where they mention the need for many tricks, such as policy learning rate rescaling, and that parameter sharing was detrimental to good empirical results, thus requiring a separate network. This separate network also necessitates a corresponding optimizer state, which by default takes twice the network memory, unless a more complex optimizer is used.
>
>
> For all these reasons, we chose to design an algorithm without target networks. Furthermore, to ensure a fair comparison, we use DRO-V (the version without target networks) from the DRO paper as a baseline.
>
>     q1) For the implementation details, during optimization, are  $\pi_l$ and $v_l$ in Eq.10 updated simultaneously? Is stop-gradient applied? If the gradient of ShiQ can be provided in the paper, it may be more convenient to understand the optimization objective.
>
> Thank you for your question. This is a very good point. Indeed, $v_l$ and $\pi_l$ are updated simultaneously, as the loss is only on the logits of the policy. Additionally, there is no stop-gradient in our loss, because it is a residual approach. Regarding the computation of the gradient, we completely agree with your point and will add the gradient computation to our paper.
>
>      q2) In the multi-turn experiments, does DRO use single-step rewards or multi-step returns? If DRO is modified to use multi-step returns, would it be equivalent to ShiQ (ignoring value modeling differences)? What would the empirical results show?
>
>
> In multi-turn experiments, DRO uses a multi-step reward, which is the sum of the rewards. This is not equivalent to ShiQ, as ShiQ uses a sum of square terms rather than a square of a sum in the multi-turn setting of DRO. Additionally, ShiQ includes a third shift at initialization that DRO does not have. Finally, we do not use target networks. For all these reasons, our approach differs from DRO.
>
>
>     q4) Minor Issue: The subscripts in Equation 9 are incorrect？
>
> Thanks for pointing this. You are totally write we will correct typos.

---

> > ### Comment · Reviewer_CDVa · 2025-08-07
> >
> > Thank you for the author's reply. The author clarified the differences from the DRO loss, noting that ShiQ outperforms DRO on multi-turn tasks, with distinctions in optimization, initialization, and the use of additional networks.
> >
> > I still have a few questions. Could the author elaborate further on DRO’s implementation? When referring to "a square of a sum," does this mean the current DRO implementation computes only a single "square of sum" for the entire multi-turn sequence?
> >
> > If we disregard initialization and auxiliary networks, and set $\gamma=1$, would ShiQ essentially be equivalent to modifying DRO by applying a square at the end of each token, where the sequence is truncated at that point and R represents the subsequent return? I want to understand where the advantage of ShiQ over DRO comes from in multi-turns task.
> >
> > $L_ {\text{DRO}_ {mt}} = \mathbb{E} \left[ \sum_ {t=1}^{n} \left( R(x, y_t) - \beta \ln \frac{\pi(y_t|x)}{\pi_ {\text{ref}}(y_t|x)} - V(x,y_t) \right)^2 \right].$

---

> > > ### Author Response · Authors · 2025-08-07
> > > **Answer to Official Comment by Reviewer CDVa**
> > >
> > > Thanks for your detailed answer and relevant questions.
> > >
> > > Regarding the DRO implementation, the loss function is detailed in Appendix D.3 (specifically, the DRO-V loss).
> > >
> > > In our implementation, there is only one squared term, positioned at the beginning of the sequence. This resembles a bandit-style algorithm but incorporates the sum of rewards over the entire trajectory, similar to the multi-turn approach in DPO implementations.
> > >
> > > The multi-turn loss for $DRO_{mt}$ , as written, lacks theoretical justification. The transition from the classical DRO loss to the proposed loss is not straightforward or directly derivable.
> > >
> > > If we disregard ShiQ's initialization concept, omit the auxiliary networks, set
> > > $\gamma$ to 1, and assume access to value networks (which is impractical without our proposed trick and no additional network), applying a square at the end of each token, then the losses would indeed be equivalent. However, this equivalence relies on four distinct conditional statements.
> > >
> > > To conclude regarding why ShiQ performs better in multi-turn settings compared to DRO, we believe that your proposed $DRO_{mt}$ loss would likely outperform our implementation of DRO as it seems more fine-grained loss. However, there is no theoretical justification for this, and we have not encountered this specific loss formulation in any existing literature.

---

> > > > ### Comment · Reviewer_CDVa · 2025-08-07
> > > >
> > > > Thank you for the clarification regarding DRO’s implementation and its distinctions from ShiQ, including the detailed formulation of the ShiQ loss.
> > > >
> > > > I still have two minor considerations.
> > > >
> > > > First, while the practical effectiveness of ShiQ is promising, it might be worth further exploration whether the heuristic $DRO_{mt}$ could achieve similar empirical performance. I acknowledge the author's concern that its theoretical support is weaker compared to ShiQ, but this could potentially diminish the contribution in terms of experimental effectiveness if a heuristic method can already handle it.
> > > >
> > > > Second, regarding the value network ablation, while I understand the computational constraints, a comparison, even at a smaller scale (existing open-source frameworks can fully support PPO training for 7B-scale models with multiple networks), could strengthen the empirical validation.
> > > >
> > > > That said, I also appreciate ShiQ’s theoretical contributions and ablations of different variants. Thus, I maintain my original positive scores.

---

### Decision · Program_Chairs · 2025-09-17

**Decision:**

Accept (poster)

**Comment:**

This paper introduces ShiQ (Shifted-Q), a new reinforcement learning framework for fine-tuning LLMs by adapting Q-learning with theoretically grounded loss functions derived from the Bellman equation. Unlike standard on-policy policy gradient methods such as PPO, ShiQ supports off-policy, token-level, and multi-turn learning, enabling sample efficiency and offline training without the need for target networks. The authors present both theoretical derivations—proving convergence and addressing initialization and memory concerns—and practical evaluations on toy tasks, single-turn benchmarks (HH, UltraFeedback), and multi-turn LLM benchmarks (BFCL-V3), showing that ShiQ can outperform or match state-of-the-art methods in alignment tasks.

The paper's strengths include a strong theoretical foundation, clear motivation for bringing Q-learning to LLMs, effective ablations demonstrating the role of each design choice, and novel adaptations like memory-friendly value modeling. However, concerns included limited large-scale empirical validation (R1, R3), closeness to prior DRO formulations (R2), notation complexity (R3), and assumptions about token-level rewards (R5).

Overall, I recommend acceptance. The paper provides a timely and well-justified contribution, reviving Q-learning for LLMs in a way that is theoretically principled and practically relevant. Despite some empirical limitations, the novelty of ShiQ lies in bridging reinforcement learning theory with LLM fine-tuning, offering perspectives beyond the current dominance of policy gradient methods. Please incorporate the final suggestions of the reviewers in the camera-ready version as promised